# The Association of the COVID-19 Pandemic with the Uptake of Colorectal Cancer Screening Varies by Socioeconomic Status in Flanders, Belgium

**DOI:** 10.3390/cancers16233983

**Published:** 2024-11-27

**Authors:** Senshuang Zheng, Lilu Ding, Marcel J. W. Greuter, Thuy Ngan Tran, Grigory Sidorenkov, Sarah Hoeck, Mathieu Goossens, Guido Van Hal, Geertruida H. de Bock

**Affiliations:** 1Department of Epidemiology, University of Groningen, University Medical Center Groningen, 9700 RB Groningen, The Netherlands; s.zheng02@umcg.nl (S.Z.);; 2Department of Epidemiology and Health Statistics, School of Public Health, Hangzhou Medical College, Hangzhou 310053, China; 3Department of Radiology, University of Groningen, University Medical Center Groningen, 9700 RB Groningen, The Netherlands; 4Family Medicine and Population Health (FAMPOP), Faculty of Medicine and Health Sciences, University of Antwerp, 2610 Antwerp, Belgium; 5Centre for Cancer Detection, 8000 Bruges, Belgiumguido.vanhal@uantwerpen.be (G.V.H.); 6Social Epidemiology and Health Policy, University of Antwerp, 2610 Antwerp, Belgium

**Keywords:** socioeconomic status, colorectal cancer, cancer screening, guideline uptake, COVID-19, Belgium

## Abstract

The large-scale colorectal cancer screening program (CRCSP) in Flanders, Belgium, was suspended twice in 2020 due to the COVID-19 pandemic. The aim of this retrospective study was to estimate the influence of the COVID-19 pandemic on CRCSP uptake and to identify the disproportionate changes in uptake rate and screening interval in areas with different socioeconomic status (SES). We found that the COVID-19 pandemic had a short-lived influence on uptake rate and screening interval. Areas with lower SES led to a greater decrease in uptake rate and areas with higher SES showed a greater increase in screening interval during the COVID-19 pandemic. These findings suggest that tailored invitations on CRCSP during the pandemic are needed for low SES areas. Timely warnings help people living in high SES areas, who might delay participation, to adhere to screening intervals.

## 1. Introduction

By 2022, colorectal cancer (CRC) was the third most common cancer worldwide, with an estimated 538,000 new cases diagnosed in European countries [1]. CRC screening is recommended to detect precursor lesions early to prevent CRC and to diagnose CRC at an early stage to contribute to lowering CRC-specific mortality rate [2,3,4]. In Europe, before large-scale screening programs were implemented, the stage distribution of CRC was 19% and 25% for Tumor–node–metastasis (TNM) stage I and stage IV during 1996–1998, respectively [5]. The five-year survival of TNM stage I is 70.3% and of stage IV is 6% [5]. Previous research showed that in screening about 50% of detected CRCs are stage I and only 6% are stage IV, which could improve CRC survival [6]. High penetration rates and responding to invitations on time are critical factors for screening programs to achieve optimal effects [7,8,9].

Due to the outbreak of the COVID-19 pandemic and over 54,000 related deaths in early 2020 in Europe, several countries implemented strict measures to slow the spread of COVID-19, including Flanders, Belgium [7,10]. The large-scale CRC screening program (CRCSP) in Flanders was suspended twice in 2020 [7]. Overall, the COVID-19 pandemic was found to have little influence on invitation coverage and uptake in Flemish CRCSP [7]. However, a systematic review indicated that CRC screening declined by 52% in Europe, which led to delayed diagnosis and an increase in advanced CRC cases [11,12,13,14]. Several studies explored the unequal changes in the pandemic on cancer screening among populations with different characteristics, yielding contradictory results. Lee et al. found that the greatest reduction in cancer screening uptake rates was shown in populations with high socioeconomic status (SES), older adult women, and those without chronic health conditions [15]. However, Fedewa et al. indicated that the low SES group had greater decreases in uptake rate [16].

Jidkova et al. found that the association of COVID-19 with uptake of Flemish screening programs varied among different age groups and screening history groups [7]. However, SES variables were not included. As low SES acts as a barrier to participation in CRC screening [17,18], it can be questioned whether this is also the case for the low SES group during the pandemic. For that, this study aimed to evaluate the differences in screening uptake rate and screening interval of the CRCSP due to the COVID-19 pandemic compared to before the pandemic, and to identify the correlation of SES on these differences. We used data from Flemish CRCSP. These results may help find possible barriers of screening in the context of a public health emergency and to improve uptake of CRCSP.

## 2. Materials and Methods

### 2.1. CRC Screening in Flanders

An organized CRCSP has been implemented since 2013 in Flanders, Belgium [19]. A biennial fecal immunochemical test (FIT) is recommended for screening-eligible individuals aged 50–74 years [20]. Individuals were excluded from CRCSP if they have had a stool test in the last two years, a virtual colonoscopy in four years, a complete colonoscopy or diagnosed with CRC in ten years, or a previous colectomy [19]. The Centre for Cancer Detection (CCD) sends invitations with leaflets, and a free-of-charge FIT every two years to all eligible citizens. An (e-)reminder is sent to non-participants after ten weeks. Participants return the completed FIT with a participation form to the laboratory by post [7,19]. Between February 2017 and September 2018, the interval between invitations was shortened to 22 months, with a 24-month interval for the rest of the period [19,21]. More than 60% of cancers detected were TNM stage I CRC and carcinoma in situ during 2013–2018 [19].

### 2.2. Interruption Period

The first interruption in CRCSP occurred between 22 March and 23 May 2020 (weeks 12–20), and the second interruption occurred between 15 November and 28 November 2020 (weeks 46–47) [7].

### 2.3. Study Design and Data Sources

We carried out a retrospective analysis of the CRCSP in Flanders, Belgium. Data of CRCSP were retrieved from the CCD database in Flanders and were aggregated at a statistical sector level to desensitize individual information and ensure anonymity. SES data were derived by linkage to the Provincies InCijfers database (https://provincies.incijfers.be/databank, accessed on 24 October 2024), using a statistical sector identification code. The statistical sector is an area smaller than a neighborhood and is the smallest geographical unit at which screening and administrative information is compiled [22]. We used the term area hereafter. The unit of analysis for all statistics was the area. There are 9494 areas in Flanders, Belgium. Areas were excluded if less than five people were screened/not screened to avoid identification of individuals.

### 2.4. Study Population

The study population included people aged 50–74 in Flanders, who received the invitation between 1 January 2018 and 30 April 2022.

### 2.5. Definitions

The primary and secondary outcomes were the difference in screening uptake rate and the difference in screening interval during and before COVID-19, respectively. The screening uptake rate was the short-term response rate, which was defined as the percentage of people returning FIT within 40 days after sending out the invitation letter. It was calculated as the number of uptake people in 40 days divided by the total number of people receiving invitations in an area. The screening interval was defined as the number of days between the current screening date (the laboratory received the returned FIT) and the previous screening date. It was calculated as the median number of days among all individuals with both current and previous screening dates in an area. The uptake rate difference and the interval difference were calculated by subtracting the overall uptake rate and the overall interval in the period before COVID-19 from those in the period during COVID-19 for each area. A negative/positive uptake rate indicated a decrease/increase in the percentage of people returning FIT within 40 days during COVID-19 compared to before COVID-19. This was similar to the interval difference.

Because CRCSP was suspended in April 2020 and restarted in May, we defined May 2020 to April 2021 as the period during COVID-19. Additionally, people are invited to participate in the CRCSP biennially, so May 2018 to April 2019 was defined as the period before COVID-19. Since the CCD adjusted the interval between invitations to 22 months from January to September 2018, we added 60 days to the interval of this period in each area.

Data on male percentage, percentage of people over 60 years old, and screening participation history rate were collected from the CCD database. The screening participation history rate refers to the proportion of individuals who had previously participated in the CRCSP among those invited to participate again, excluding those invited for the first time.

The SES variables were extracted for each area in 2020: (1) population density: the number of inhabitants per square kilometer; (2) same address as last year: the percentage of inhabitants with the same address as last year; (3) Belgian origin nationality: the percentage of inhabitants with a Belgian nationality at birth; (4) living alone: the percentage of inhabitants living alone; (5) children in single-parent families: the percentage of children aged 0–17 living in single-parent households among all inhabitants; (6) unemployed job seekers: the percentage of inhabitants unemployed and seeking job; (7) average net taxable income per inhabitant: the quotient of total net taxable income and total number of inhabitants; (8) home ownership: the percentage of inhabitants having their own house; (9) social housing per 100 private households: the quotient of total number of social housing and total number of private households; (10) cars per 100 households: the quotient of total number of cars and total number of households.

### 2.6. Statistical Analysis

Monthly screening uptake rate and screening interval were reported as median and interquartile ranges from 2018 to 2022, and as mean and 95% confidence intervals (CI) by quartile of SES variables. We applied quantile regression as the distribution ranges of difference in screening uptake rate and difference in screening interval increased with the increase or decrease in all independent variables (Figure 1 and Appendix A). Quantile regression used the weighted least square to estimate the central tendency and determinants of the outcomes at different levels. For that, areas were ranked in ascending order by the difference values in uptake rate and by the difference values in screening interval, respectively, and categorized into 10 quantiles (Q10–Q90), with Q10 representing the areas with the smallest 10% of values. Both outcomes and SES variables were described using the median per quantile. Univariate regression was used to select variables for the multivariable model and 10-quantile fit curves were obtained for each variable. Multicollinearity was checked among the included variables, and those with variance inflation factors >10 were excluded. Multivariable quantile regression was applied to assess coefficients and 95% CIs. The male percentage, percentage of people over 60 years old, and screening participation history rate were included as potential confounders. A *p*-value of <0.05 was considered as a statistically significant difference. Statistical analysis was performed using R version 4.3.1.

## 3. Results

### 3.1. CRCSP Screening Uptake Rate and Screening Interval in 2018–2022

During 2020, the CRCSP uptake stopped in April. The uptake rate reached the lowest value of 22.2% in March. After the CRCSP restarted in May, the rate bottomed out at 26.3% in August and gradually increased back to the rate before the interruption. The uptake rate decreased significantly in March compared to that in the same months of 2018 and 2019 (40.0% and 42.9%). The median screening interval peaked at 819.0 days in May, then narrowed to a shorter interval of 762.0 days in August and grew until December. (Appendix A). Stratifying the data by SES variables, areas in the highest quartile for the percentage of people living alone, children in single-parent families, and unemployed job seekers, and those in the lowest quartile for average income, showed lower uptake rates during 2018–2022 (Appendix A). Areas in the lowest quartile for the percentage of people living alone, children in single-parent families, and unemployed job seekers, and those in the highest quartile for average income, showed greater screening intervals (Appendix A).

### 3.2. Description of SES Variables and Outcomes

There were 8382 areas with uptake rate difference and 8270 with screening interval difference. The median uptake rate differences were −15.1% in Q10 and 5.5% in Q90. The lowest values for population density (310.0/km^2^), percentage of living alone (9.9%), and percentage of children in single-parent families (4.6%) were observed in Q10. The highest value for the percentage of home ownership (84.6%) was found in Q10. Higher average incomes were found in Q10 and Q90 (Table 1).

The median screening interval difference increased from −6.0 days in Q10 to 20.0 days in Q90. In Q90, the percentage of those living alone (10.0%) and the percentage of unemployed job seekers (3.5%) presented lower values, and average income (€21,280.0) and percentage of home ownership (84.1%) showed the highest values (Table 2).

### 3.3. Determinants of Differences in Screening Uptake Rate

In Q30–Q80 of the difference in screening uptake rate, areas with lower average income led to smaller difference values in uptake rate, indicating greater decreases in screening uptake rate during COVID-19 compared to before COVID-19. The coefficients enlarged as the uptake rate difference value increased. Areas with higher percentages of home ownership in Q20–Q60 and higher percentages of living alone in Q20–Q80 showed a greater decrease in screening uptake rate. Areas with lower population density showed a greater decrease in screening uptake rate in Q10–Q50 and a greater increase in Q80–Q90. The coefficients were 0.07 to 0.10 in Q30–Q80 for average income, −0.09 in Q20 for living alone, −0.06 in Q20 for home ownership, 0.43 in Q10, and −0.29 in Q90 for population density (Figure 2 and Appendix A).

### 3.4. Determinants of Difference in Screening Interval

In Q10 of the difference in screening interval, a lower population density, a higher percentage of living alone, unemployed job seekers, and home ownership, and a greater number of cars were associated with a greater decrease in screening interval. A greater increase in screening interval was associated with an increased average income in Q20–Q90 and a decreased population density in Q50–Q90. The correlation between average income and population density enhanced with an increase in interval difference. The associations in opposite directions were observed in Q10–Q20 versus in Q50–Q90 for population density. The coefficients were 0.31 for population density, −0.13 for living alone, −0.32 for unemployed job seekers, −0.05 for home ownership, −0.05 for number of cars in Q10, 0.42 in Q90 for average income, and −0.75 in Q90 for population density (Figure 3 and Appendix A).

## 4. Discussion

CRCSP screening uptake rate decreased in March 2020 and screening interval increased between May and July 2020 compared to what was observed before the COVID-19 pandemic. Areas with more people living alone, lower income, and higher rates of home ownership may lead to a greater decrease in screening uptake rate. Lower population density and higher income were determinants of a greater increase in screening interval. In addition, the amplitudes of difference in uptake rate and screening interval decreased as population density increased. The correlations of income with uptake rate and screening interval differences enhance as the values of both outcomes increase.

Our results showed that the COVID-19 pandemic had little influence on Flemish CRCSP. Screening uptake rate decreased during the first few months of the pandemic in early 2020, and there was a corresponding increase in screening interval when the CRCSP restarted after an eight-week interruption. These findings are consistent with a previous study on the Flemish CRCSP [7]. The reduction in uptake rate and the extension of the screening interval might be caused by a backlog of invitations during the CRCSP interruption [7]. A study from the US also presented that the CRC screening uptake rate dropped significantly from March to May 2020, and it went back to a similar level to that before COVID-19 by July [23]. Another study of cervical cancer screening showed a 25% reduction in participation between June and September 2020. After the stay-at-home order was lifted, it returned to pre-pandemic levels [24]. These studies indicated a decline in the use of cancer screening at the beginning of the pandemic, but the change was not long-lasting [25]. This may be explained by the fact that the FIT is a simple test to be performed by invitees themselves, and returned by post, without making any appointment [7,19]. Thus, the influence of COVID-19 on uptake is limited. This may be attributed to the measures implemented to prevent a decline in the utilization of prevention services during the COVID-19 pandemic, such as a nationwide media campaign to promote the use of healthcare services in Belgium. In the US, screening reminder systems and tracking persons lost to follow-up were implemented [7,24].

Our main finding was that more people living alone, and lower income were determinants of a greater decrease in screening uptake rate, and a smaller number of cars was the determinant of a greater increase in screening interval during the COVID-19 pandemic. These characteristics represent lower SES. This association is similar to what Miller et al. and Fedewa et al. found that people living in poverty areas or with low SES were less likely to attend cancer screening during COVID-19 [16,26]. A possible explanation is that people with low SES may have less health insurance coverage and less access to high-quality healthcare, which are key barriers to delivering effective healthcare [27,28,29]. Moreover, during the pandemic, the population with low SES might be more concerned about basic living needs, while putting healthcare services on the back burner [30]. Reducing SES disparities in health behaviors requires government intervention [31]. The disparities could be reduced by raising public awareness of cancer prevention by media publicity and one-on-one health education, and by improving screening reminder systems [32]. Providing tailored, detailed information about the free-of-charge FIT, and the benefits and safety of screening during the pandemic, specifically targeting low SES areas, might be effective in reducing disparities in CRCSP uptake.

Opposite associations were shown in the quantiles with negative difference values in uptake rate versus those with positive values for population density, and a larger amplitude of difference is more likely to occur in areas with lower population density. Similar associations were found for screening interval differences. Urbanization is the main factor driving changes in population concentration [33]. The relationship between SES and population density cannot be generalized. In Western European countries, areas with a high level of urbanization are likely to have a higher population density [34]. One possible explanation for the greater decrease in screening uptake rate and the greater increase in screening interval is that areas with low population density may be semi-rural areas in Flanders, where residents could have less education and awareness on cancer prevention. This contributes to lower uptake rates and longer screening intervals [35,36]. Another reason might be that the amount of eligible people is high in higher-density areas. The sheer number of eligible residents leads to a more stable uptake rate and screening interval than those in low-density areas, where changes in behavior among a smaller number of residents might have more noticeable changes in outcome.

We observed a positive correlation between income and an increase in screening intervals during the pandemic. Fewer people living alone, and lower unemployment were associated with a greater increase in screening interval. A higher percentage of home ownership was associated with a greater decrease in screening uptake rate. These indicators represent higher SES. This finding is consistent with the study of Chen et al., which found that the decline in uptake rate was largest among individuals with the highest SES in 2020 [23]. World Health Organization indicates that people with higher income and higher SES generally keep healthy lifestyles [37]. Additionally, people with higher incomes may have better access to healthcare services, allowing them to decide if and when to participate in the CRCSP [38,39]. Limited free time might be another barrier to participation. Thus, the large-scale CRCSP might not be a priority for high SES populations, resulting in non-participation or postponed participation. Furthermore, our data indicated that high-income areas have a higher proportion of people aged 60 and over. Older people may delay participating in screening due to health problems or fear of disease [18,40,41]. This study also showed that areas with high unemployment rates have a smaller increase in screening intervals. Unemployed people may have more free time and the anxiety caused by unemployment, which prompts people to pay more attention to their health and participate in free CRCSP on time [42]. It is important to note that the unemployment rate during the COVID-19 pandemic rose to levels not seen since the Great Depression [43]. Job loss leads to both reductions in income and access to employer-based health insurance coverage, which is a barrier to receiving preventive care.

From our findings and results of prior studies, we can conclude that SES and social environment factors play a critical role in cancer screening uptake and inequality in access to health services. The influencing factors include awareness, affordability of healthcare, and external support. People with disadvantageous socioeconomic conditions may have inadequate knowledge of screening, leading them to perceive screening as lacking benefit and effectiveness and as an unnecessary service [36,44,45]. Personal financial concerns and worries about the cost of post-detection treatment can also pose barriers to participation [29]. Therefore, addressing barriers to screening participation from a socioeconomic and environmental perspective will have the potential to improve screening behaviors among populations.

### Strengths and Limitations

Our findings on the association of the COVID-19 pandemic with CRCSP support the results of Jidkova et al. [7]. Additionally, this study incorporated SES variables to be the first to explore the disproportionate association of the pandemic with the uptake of the Flemish CRCSP across different SES populations, providing a more comprehensive perspective on the barriers to screening behaviors. The strength of this study was that the validity of all data is warranted by the official registration database, which avoids recall and reporting bias. There are several limitations. First, all data were aggregated at the statistical sector (area) level due to individual privacy considerations. Area-level data might not accurately represent individual status, and the misclassification may weaken the true association between variables and changes in screening uptake. Therefore, generalizations of the results should be made with caution. Nonetheless, it is a smaller unit than the neighborhood. This means that the characteristics of residents in an area were presented more accurately than those in a neighborhood. Second, since the data were retrieved from two databases, we could only use the SES characteristics of all residents in an area to represent those of the residents who were invited to CRCSP. This might introduce classification bias. However, there are geographic variations in healthcare utilization, and residents living in the same neighborhood show similar SES and health behaviors [46,47]. Thus, the impact of this limitation could be mitigated. Third, we did not include education level and occupation type because the smallest aggregation unit for these indicators was at the municipality level. With 9494 areas in 300 municipalities, the variations could not be presented among the areas within each municipality. In further studies, SES should be comprehensively evaluated by obtaining more SES indicators to explore the correlation between SES and screening behavior. Fourth, housing quality and neighborhood characteristics may better reflect measures of SES, but these variables were excluded in this study due to the lack of data at the area level. However, social housing, which was extracted as an SES variable at the area level, was included in regressions and is a good indicator of SES.

## 5. Conclusions

During the COVID-19 pandemic, there was an overall short-lived change in the screening uptake rate and screening interval due to the interruption of CRCSP in Flanders. People in lower SES areas, indicated by higher percentages of living alone and lower income, showed a greater decrease in screening uptake rate. A higher SES population, characterized by higher income, fewer people living alone, and lower unemployment, showed a greater increase in screening interval. Tailored invitations with detailed information on the benefits and safety of CRCSP during a pandemic are essential for increasing uptake among lower SES populations. A timely warning may help people with high SES, who delay participation, adhere to screening intervals, especially during public health emergencies.

## Figures and Tables

**Figure 1 cancers-16-03983-f001:**
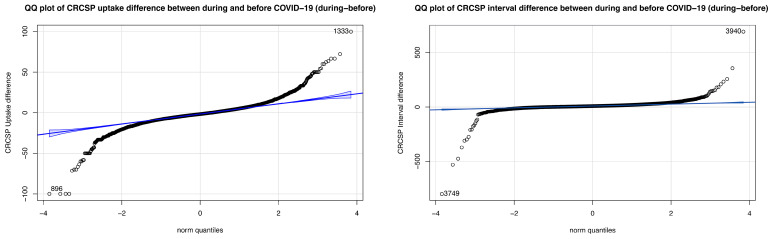
Normality test of difference in uptake rate and difference in screening interval of colorectal cancer screening program between the period during COVID-19 and the period before COVID-19 among statistical sectors in Flanders, Belgium. Footnotes: CRCSP: Colorectal cancer screening program. Each circle symbol represents a single data point from the dataset. Blue lines and bands are the ideal normal distribution and confidence bands.

**Figure 2 cancers-16-03983-f002:**
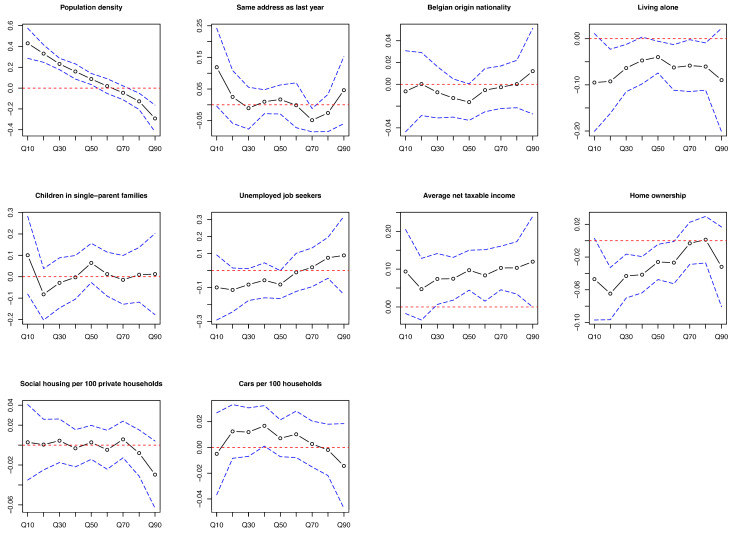
Coefficients of multivariable quantile regression of all variables in each quantile of difference in uptake rate of colorectal cancer screening program between the period during COVID-19 and the period before COVID-19 among statistical sectors in Flanders, Belgium. Footnotes: The black symbol lines are coefficients of each quantile. The blue dotted lines are the 95% confidence interval. The red dotted lines are the 0 lines. Q10: Areas with the smallest 10% of uptake difference values; Q90: Areas with the highest 10% of uptake difference values.

**Figure 3 cancers-16-03983-f003:**
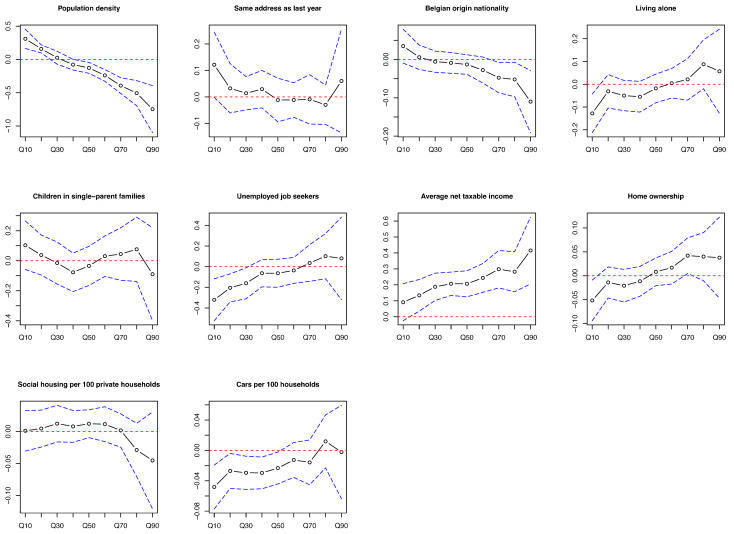
Coefficients of multivariable quantile regression of all variables in each quantile of difference in screening interval of colorectal cancer screening program between the period during COVID-19 and the period before COVID-19 among statistical sectors in Flanders, Belgium. Footnotes: The black symbol lines are coefficients of each quantile. The blue dotted lines are the 95% confidence interval. The red dotted lines are the 0 lines. Q10: Areas with the smallest 10% of uptake difference values; Q90: Areas with the highest 10% of uptake difference values.

**Table 1 cancers-16-03983-t001:** Description of variables in 2020 at the 10th (Q10) to the 90th (Q90) quantiles of difference in uptake rate of colorectal cancer screening program among statistical sectors in Flanders, Belgium.

Variables/Outcome	Q10	Q20	Q30	Q40	Q50	Q60	Q70	Q80	Q90
Median uptake difference, %	−15.1	−8.4	−5.7	−3.8	−2.3	−0.9	0.6	2.7	5.5
Uptake difference range, %	≤−10.8	(−10.8, −7.0)	(−7.0, −4.8)	(−4.8, −3.1)	(−3.1, −1.7)	(−1.7, −0.1)	(−0.1, 1.7)	(1.7, 3.8)	(3.8, 7.6)
Male percentage, %	51.2	51.3	51.0	50.5	50.7	50.6	50.4	50.7	51.0
≥60 years old percentage, %	47.1	48.1	49.0	49.4	49.3	49.0	48.7	48.6	49.2
Participation history rate, %	55.6	54.7	54.8	53.6	54.4	53.9	54.1	53.9	54.6
Population density, /km^2^	310.0	1100.0	1260.0	1530.0	1440.0	1590.0	1490.0	1290.0	1100.0
Same address as last year, %	93.2	93.1	92.7	92.4	92.4	92.1	92.2	92.3	92.7
Belgian origin nationality, %	93.1	92.5	91.7	91.0	91.0	90.2	90.6	91.7	92.5
Living alone, %	9.9	10.2	10.3	11.1	10.5	11.1	11.0	10.5	10.1
Children in single-parent families, %	4.6	4.8	4.9	5.1	5.0	5.3	5.0	4.9	4.9
Unemployed job seekers, %	3.4	3.5	3.7	3.8	3.7	3.9	3.9	3.7	3.6
Average net taxable income per inhabitant, €	21,080.0	21,050.0	20,670.0	20,560.0	20,660.0	20,510.0	20,570.0	21,170.0	21,190.0
Home ownership, %	84.6	83.3	81.0	79.2	80.1	78.0	78.5	81.3	83.0
Social housing, per 100 private households	0.0	0.0	0.2	0.4	0.2	0.6	0.3	0.0	0.0
Cars, per 100 households	134.8	132.3	129.8	127.8	129.8	127.2	127.8	130.8	133.9

Footnotes: Q10: Areas with the smallest 10% of uptake difference values; Q90: Areas with the highest 10% of uptake difference values.

**Table 2 cancers-16-03983-t002:** Description of variables in 2020 at the 10th (Q10) to the 90th (Q90) quantiles of difference in screening interval of colorectal cancer screening program among statistical sectors in Flanders, Belgium.

Variables/Outcome	Q10	Q20	Q30	Q40	Q50	Q60	Q70	Q80	Q90
Median screening interval difference, days	−6.0	1.0	3.5	6.0	7.5	9.5	12.0	15.0	20.0
Interval difference range, days	≤−1.5	(−1.5, 2.0)	(2.0, 4.5)	(4.5, 6.5)	(6.5, 8.5)	(8.5, 10.5)	(10.5, 13.0)	(13.0, 17.0)	(17.0, 24.0)
Male percentage, %	50.8	51.1	50.4	50.4	50.6	50.5	51.1	51.1	51.2
≥60 years old percentage, %	49.2	48.9	50.0	50.5	49.3	48.8	48.7	48.0	47.3
Participation history rate, %	52.9	53.9	56.0	54.5	55.0	56.4	54.6	54.3	53.9
Population density, /km^2^	480.0	1250.0	1450.0	1620.0	1490.0	1420.0	1330.0	1160.0	940.0
Same address as last year, %	93.0	92.6	92.6	92.3	92.4	92.2	92.4	92.7	93.1
Belgian origin nationality, %	93.0	91.5	91.1	90.9	90.7	91.8	91.6	91.9	92.6
Living alone, %	9.9	10.4	10.6	11.1	10.7	10.6	10.6	10.1	10.0
Children in single-parent families, %	4.7	4.8	5.0	5.0	5.1	4.8	5.0	4.8	4.8
Unemployed job seekers, %	3.6	3.7	3.9	3.9	3.8	3.6	3.7	3.5	3.5
Average net taxable income per inhabitant, €	20,900.0	20,790.0	20,620.0	20,660.0	20,450.0	20,810.0	20,920.0	21,130.0	21,280.0
Home ownership, %	83.9	81.9	81.3	79.7	79.0	79.7	79.8	82.5	84.1
Social housing, per 100 private households	0.0	0.0	0.3	0.4	0.4	0.3	0.3	0.0	0.0
Cars, per 100 households	134.6	130.9	130.0	127.8	128.6	129.5	129.7	132.0	133.6

Footnotes: Q10: Areas with the smallest 10% of uptake difference values; Q90: Areas with the highest 10% of uptake difference values.

## Data Availability

Data are unavailable due to privacy or ethical restrictions.

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
