# Peer review of "The Association of the COVID-19 Pandemic with the Uptake of Colorectal Cancer Screening Varies by Socioeconomic Status in Flanders, Belgium"

_cancers, 2024, doi:10.3390/cancers16233983_

Round 1

Reviewer 1 Report

Comments and Suggestions for Authors

The manuscript can be accepted in present form. 

Figure S1 (normality test) from the supplementary files should be included in the manuscript

Author Response

Dear reviewer,

We appreciate your valuable comments and suggestions and have revised the manuscript accordingly, highlighting all changes using the "Track Changes" (in red) function in the file of “Main text-revised version”. Details of the revisions in the manuscript and our response to the comments are shown bold in this rebuttal letter.

Comments 1: The manuscript can be accepted in present form. 

Figure S1 (normality test) from the supplementary files should be included in the manuscript

Response 1: Thank you for your suggestion. We have moved the Figure S1 to the main text on Pg. 4, lines 174-178. We have also revised the numbering of the figures in the manuscript and supplementary materials accordingly.

Reviewer 2 Report

Comments and Suggestions for Authors

The manuscript is clear, relevant for the field and addresses a pertinent issue regarding how the COVID-19 pandemic affected the colorectal cancer screening. The article follows a well-structured format and the methodology is scientifically sound. The results and the conclusions are well addressed.

The study includes recent studies that justify the results of the authors. The references are appropriate for supporting the authors’ findings.

The study uses a retrospective approach which is appropriate to test the author’s initial hypothesis.

The manuscripts results are reproducible based on the details given in the methods section. The quartile-based analysis allows other researchers to replicate the approach to similar populations.

The figures are appropriate and effectively provide the data regarding the screening intervals by socioeconomic stratification.

The statistical analysis addresses the observed effects in a scientifically sound manner.

The conclusions are consistent the evidence and arguments presented. The authors emphasize that SES factors had significant impact on the screening uptake during the COVID-19 pandemic.

The authors mention the anonymization and privacy of the data which a suitable feature for studies that involve sensitive health data.

The study addresses a gap by exploring the socioeconomic disparities and their impact on colorectal cancer screening during the pandemic. Moreover, the article provides a valuable addition to public health regarding the effects of the pandemic on the preventive healthcare settings. While previous studies have investigated the effect of COVID-19 on the screening of cancer in general, this study’s focus is on colorectal cancer screening in Flanders area which is the gap that comes to fill.

Author Response

Dear reviewer,

We appreciate your valuable comments and nice words.

On behalf of all co-authors,

Yours sincerely,

Senshuang Zheng, Marcel J. W. Greuter, Geertruida H. de Bock

Reviewer 3 Report

Comments and Suggestions for Authors

The manuscript is interesting, although the results are derived from aggregated data rather than individual-level data on screening adherence. This could introduce a potential misclassification bias. How it might affect the findings?

Additionally, I suggest emphasizing the role of social determinants in influencing screening uptake and addressing related inequalities in access. Social determinants play a crucial role and should be discussed to provide a more comprehensive perspective on barriers to screening.

Furthermore, I noticed that over 15% of the references are self-citations. While this is not inherently problematic, it may raise concerns about the diversity and balance of the cited literature. Consider to remove some self-citations. Expanding the references to include a broader range of independent studies would enhance the manuscript's credibility and provide a more robust evidence base.

Author Response

Dear reviewer,

We appreciate your valuable comments and suggestions and have revised the manuscript accordingly, highlighting all changes using the "Track Changes" (in red) function in the file of “Main text-revised version”. Details of the revisions in the manuscript and our response to the comments are shown bold in this rebuttal letter.

Comments 1: The manuscript is interesting, although the results are derived from aggregated data rather than individual-level data on screening adherence. This could introduce a potential misclassification bias. How it might affect the findings?

Response 1: Thank you for your comments. Indeed, using aggregated data instead of individual-level data might introduce misclassification bias, and it could affect the findings. We have added this to the first limitation on Pg. 10, paragraph 3, lines 349-352: “Area level data might not accurately represent individual status, and the misclassification may weaken the true association between variables and changes in screening uptake. Therefore, generalization of the results should be made with caution.

Comments 2: Additionally, I suggest emphasizing the role of social determinants in influencing screening uptake and addressing related inequalities in access. Social determinants play a crucial role and should be discussed to provide a more comprehensive perspective on barriers to screening.

Response 2: Agree. We have, accordingly, added the following text to the last paragraph of Discussion on Pg. 10, paragraph 2, lines 331-340 to emphasize this point: “From our findings and results of prior studies, we can conclude that SES and social environment factors play a critical role in cancer screening uptake and inequality in access to health services. The influencing factors include awareness, affordability of healthcare, and external support. People with disadvantageous socioeconomic conditions may have inadequate knowledge of screening, leading them to perceive screening as lacking benefit and effectiveness and as an unnecessary service [36,44,45]. Personal financial concerns and worries about the cost of post-detection treatment can also pose barriers to participation [29]. Therefore, addressing barriers to screening participation from a socioeconomic and environmental perspective will have the potential to improve screening behaviors among populations.

Comments 3: I noticed that over 15% of the references are self-citations. While this is not inherently problematic, it may raise concerns about the diversity and balance of the cited literature. Consider to remove some self-citations. Expanding the references to include a broader range of independent studies would enhance the manuscript's credibility and provide a more robust evidence base.

Response 3: We only keep the references needed to introduce CRCSP in Flanders on Introduction and Methods and removed 2 of the self-references that provide similar information. Additionally, references have been added to the discussion, and now there are 47 references in total. The self-citation percentage now is 10%.
